# New Product Development of a Robotic Soldering Cell Using Lean Manufacturing Methodology

Emanuela Pop [1,*] , Emilia Campean [1], Ion Cristian Braga [2] and Darius Ispas [1]

1   Department of Design Engineering and Robotics, Technical University of Cluj-Napoca, Memorandumului Street, No. 28, 400114 Cluj-Napoca, Romania
2   2Connect Romania SRL, Muncii Blvd., No. 12, 400641 Cluj-Napoca, Romania
*   Correspondence: emanuela.pop@muri.utcluj.ro; Tel.: +40-742-928340

**Abstract:** With the advent of manufacturing in Industry 4.0 and consumer demand, there has been a trend of mass customization of products. This customization requirement can only be achieved through the flexibility of manufacturing processes that are tailored to meet the quality standards of customers and the large volume of production in a short time. The increase of the production capacity is achieved through the processes of industrial automation of the manufacture which maintains the increased efficiency for the series production. This study was based on the Design for Six Sigma methodology (DMADV—Define, Measure, Analyze, Design and Verify) in order to determine the soldering process characteristics and how the soldering process can be automatized. When planning the implementation of a collaborative robot in a workstation in the production plant, the following must be taken into account: steps in operations that require the most time for the worker and/or that represent factors of physical and moral overload for him; the use of adequate precision fixing devices, delimitation of work areas, sensors as well as spaces for connecting the workstation to the electrical, hydraulic/pneumatic network and constant cycle time. The proposed solution can improve the productivity of the process by integrating advanced robotics and smart devices into the soldering line.

**Keywords:** soldering; robotics; Industry 4.0; Six Sigma; automatization

## 1. Introduction

Industry 4.0 focuses on increasing the production diversity with direct influence on the manufacturing process from a technological and organizational point of view by integrating artificial intelligence, advanced robotics or cloud computing. In this way, it presented more and more intense preoccupations of the companies in the industrial field, for the development of a new requirement of the manufacture, namely the flexibility. In addition, the developed processes or products have the advantages of gaining new features such as: build-in capabilities, efficiency, cost effectives, optimization or adaptation to the actual status.

For a company to meet this requirement, it must implement flexible automation of production processes to develop and effectively produce goods with a dynamic customer-oriented evolution and alignment with modern technical and functional goals, such as Industry 4.0. [1]

Industry 4.0 has also improved the Quality System [2] and the product quality. In [2], the author proposes a framework that identifies trends using the Latent Semantic Analysis. Data from Industry 4.0 and vQM (virtual Quality Management) have been analyzed and the knowledge structures (frequency, position in text and causal relation of associated notions) of the two concepts (Industry 4.0 and vQM) have been used to identify new technological solutions. This might be used by all companies for improving the management of the companies and the quality system.

The product quality can also be improved even from the design process by using digital solutions that can provide the best solution for the product. In addition, the use of

automation, sensor or artificial intelligence can reduce the number of scrapes and detect in real time the problems that appear in the production line. In this way, an increase of productivity can be observed at companies that implemented Industry 4.0. According to [3], implementing Industry 4.0 leads to an improved productivity with 20% and a reduced manufacturing defect with 18%. Mordor intelligence [4] established that the Industry 4.0 market will grow from USD 86.03 billion in 2020 to USD 267.01 billion by 2026. Adopting new digital technologies will lead to modernized and automated processes improving the performance of the company in this way.

Human–robot integration is the next step in obtaining smart factories. A robot is defined as an "actuated mechanism programmable in two or more axes with a degree of autonomy, moving within its environment, to perform intended tasks" [5]. They were integrated within the industrial processes starting with the 1960s [6] and have risen considerably, achieving the number of 3,014,879 units in 2020 [7]. The development of digital technologies and of smart devices leads to the development of a smarter, autonomous, flexible and cheaper robot.

The purpose of this paper is to integrate collaborative robots (also cobots) in the manufacturing of electrical/electronic cables to improve production both in terms of efficiency and the need for readjustment according to the trend in this industry.

Cobots have been used before, but in industries related to manufacturing [8–11], medical sector [12–15] or even in domains such as cognitive architectures [16–18]. The problem of integrating cobots into the soldering process was not studied yet.

The term collaborative robots (cobots) refers to "a device and method for direct physical interaction between a person and a computerized manipulator" [19]. Over the last years, there has been a trend to automatize and integrate them into production lines. Currently, cobots are usually used in the industrial and manufacturing industries [20]. They can be used in "pick and place" operations, assembly, welding or even production facilities. One of the purposes of this research is to see if it is possible to integrate a cobot into the soldering process. Although cobots are used and validated in many sectors, studies regarding their appliance on soldering process are lacking. An integrated solution that can increase the overall production, be efficient and qualitative of both manual or automated soldering processes can be very useful and valuable for the companies.

A complex, qualitative and stable process is influenced by its level of automatization. The paper [21] presents the benefit of adopting cobots into manufacturing companies and, using conceptual frameworks, 39 factors were found that strengthen the adoption of them. Comparing these factors and processes with the one described in this research paper, it can seen that automation must be achieved and that most of the problems that appear in the soldering process have been also identified by the author of this paper: compatibility, complexity, velocity, flexibility, etc., to perform tasks.

In [22], the state of the art in wire-harness assembly is presented. The authors present the wire harness design process, the production process and the level of automation and innovation in this field. The author remarks that in this moment, there is no equipment that can automate the wire harness process. It is performed manually, without productivity and lots of waste and non-value-adding processes. They state that the automation of such processes needs to be taken into consideration.

In order to improve any manufacturer process and automate them, several factors need to be analyzed [23,24]: quality, cost efficiency, flexibility, ergonomics, etc. An identifying factor that influences the acceptance/adoption of cobots into manufacturing processes is the subject of [25,26]. Factors such as strategy communication, climate of trust and confidence or efficient human resource development have been studied within the study. In order to identify the factors that influence any manager in adopting cobots in process flow, several concepts have been studied: Lean Six Sigma, House of Quality or Industry 4.0 [27–32].

The House of Quality methodology was used in many sectors of the industry with the role of understanding the requirements of the customer and to find ways of improving it. The method is presented in [33] and having the role of identifying the area that can

be improved. The company's priorities must be known; however, the manufacturing decision categories must also be defined, together with the manufacturing decision categories connection. In this situation, the strategic projects that resulted for improving are: implementing lean manufacturing for a project, introducing a zero defects program or developing an area of technological processes.

Another study that analyzes the possibility of replacing manual soldering with an automated process is [34]. The authors propose an automated system with the role of conducting flexible and repeatable experiments of the UAS (Ultrasonic-Assisted Soldering) process. The parameters that were taken into consideration when designing the system are: speed of the solder tip, power, extrusion rate of the solder and the distance of the tip from the substrate. These parameters were also used for the design of the soldering device studied in this article.

The system was developed by modifying a benchtop model FDM machine base. In order to complete the automatization, a rotational control of the ultrasonic stack, a volume control of the solder and a profilometer was added. The volume control of solder allows the use of a 3000 rpm speed, the working area is situated between 250–400 mm, while the rotational control provides longitudinal vibrations to the solder tip. The used frequencies are $60 \pm 5$ kHz. In this way, a Z-axis rotational control is possible. In order to control the guidance of the solder flow to the soldering tip, a solder wire extruder subsystem was mounted. The UAS system is controlled due to the profilometer that is connected to an Arduino board.

The new automated system is more reliable and stable. Tests have shown that the solder tip is in direct relation with surface roughness, while the solder width is inversely proportional to the increasing of the solder tip speed.

In [35], the authors proposed a new automated system for the soldering of multi-wire cable, including the pre-preparing steps of the wire. The authors divided the process into the three main steps needed to obtain the cable: wire separating, wire sorting and wire pressing. All three steps have been automated, and the transition between the modules being operated by operators. The differences between the wire presented in paper [35] and the one proposed in this research is the number of connectors (37 in the case of 2Connect and 4 in the case of [35]). A camera is used to capture the status of the wire. All images are sent to a deep vision network having the role of detecting the color and locations of the wires and using the AMRWD (Attention Mask R-CNN Wire Detector) algorithm, the four colors are detected. Having this in mind, the studied process was also divided into major operations. It studies whether the soldering process can be automated, but it does not analyze which of the operation must be automated.

Another paper that studies the soldering process is [36]. The paper highlights the differences between manual and automated soldering for a PCBA board. A solder iron is used for the operation. The heat of the PCBA was limited to 120 °C due to the overheating the board. The authors used Design of Experiments to conclude the tests. The results showed that the temperature of 100 °C is optimum for obtaining the required barrel fill performance. A higher preheated temperature and a lower delta temperature between the liquid solder and the PCBA surface generates an optimum heat transfer.

From this desideratum, in order to establish the areas of interest, the process flow and the product portfolio were analyzed. This was a first challenge, because the type of production is "high-mix, low-volume", i.e., many types of parts manufactured in small series at irregular intervals. Once the operation for optimization was identified, the next major problem was the design of a workstation that would meet two apparently contradictory characteristics: high productivity with high flexibility of the production system.

An Important aspect to mention is the fact that, in the sources of information provided during the writing of this paper, no similar examples regarding automatization of the soldering process for a complex wire were found in other companies or scientific paper. This gives reason for the chosen method of planning a conceptual development; however, it also allows for the design of the workstation to involve a mainly developmental and

research approach. Mainly, for this research, we used Scopus and WoS as databases for article analysis. Taking the DMADV methodology analyzed within the literature review, the steps needed to see if the automatization of the process is needed have been presented.

In other words, the design stages of a workstation concept with collaborative soldering robot, devices and other necessary elements, ending with the approach of technological manufacturing and verification of functionality for some active elements from the final assembly, will be presented.

Due to the fact that the production of electrical/electronic cables is a dynamic market in direct connection with different sectors of the industry, which in turn are influenced by digitization and technological growth, the need for rapid, continuous improvement solutions for flexible manufacturing in this company are mandatory; all the more, most of the technological flow is manual manufacturing performed by human operators.

## 2. The Lean Manufacturing Approach for the Development of A Robotic Soldering Cell

One of the core values of the 2Connect Romania, Cluj-Napoca plant is the flexibility, and the production manufacturing model is based on HMLV (High-Mix, Low-Volume). Currently, the manufacturing process developed in 2Connect Romania is based on different process steps such as: Cutting, Stripping, Crimping, Soldering, Assembly, Potting and Over-molding.

Part of these process steps are organized as a dedicated cell or work center and part of them can be mixed in the assembly process step.

Therefore, the Over-molding process step is a dedicated cell where the Inputs/Outputs are clearly established. As the Over-molding process step is present in both facilities, in one facility the cell is organized with all machines in the "O" model, in the other facility is organized in the "L" model (as can be seen in the Figure 1).

The implementing of the Lean Manufacturing methodology on these process steps is not an important challenge as those cells usual meet in the industry, and tools such as 5S, SMED (Single Minutes Exchange Die), OEE (Overall Equipment Effectiveness), error-proofing, one-piece flow or VSM (Value Stream Mapping) could improve the processes.

The major challenge and the aim of this research paper is to improve the processes of soldering in assembly areas, mainly when the process layout is based on Job Shop and not on assembly line. The strategy to implement Lean Manufacturing methodology was previously stated as a challenge by R.M. Mahoney, who mentioned in his book "it's impossible to balance capacity in a high-mix, low-volume manufacturing facility, where the work times for each product are unequal" [37].

In addition, as mentioned by W. Laosiritaworn et. al, the Job Shop process [38] is often used when the flexibility and small batch mixing is required in order to reach the customers' demands. As a consequence, the layout of the facility in the assembly area is based on the four lines with ten Job Shops. One of the Job Shops is used often to manufacture the products that use the multi-wires cables and the named connectors D-Sub and the layout is presented in Figure 2.

Partially, the concepts used in the Lean Manufacturing have the applicability at Job Shop HMLV as well: the foundation stone 5S issued initially by Taiichi Ohno (who developed the Toyota Production System) is the first needed and easily implemented tool to visualize the process and to generate the environment for the next Lean tools [39]. The standardized work was also the other Lean method implemented in 2Connect and this brought added value.

However, the customer's needs and production planning still require improvements such as "one-piece flow" and "line balancing" in order to obtain the quick reaction and data for the analysis. Those two Lean methods cannot be appropriate for the Job Shop in HMLV; therefore, the soldering of D-Sub connectors is only possible by implementing "employee involvement", which is most critical in the creation of the working environment with added value or win–win relationships between employees, organization [40] and "top-down leadership".

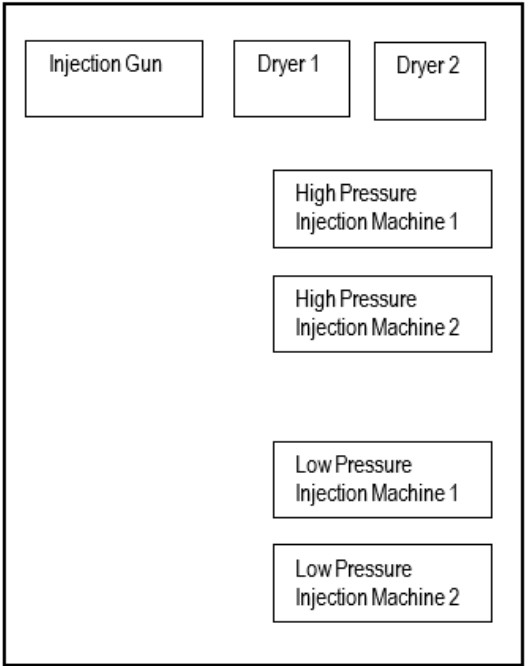

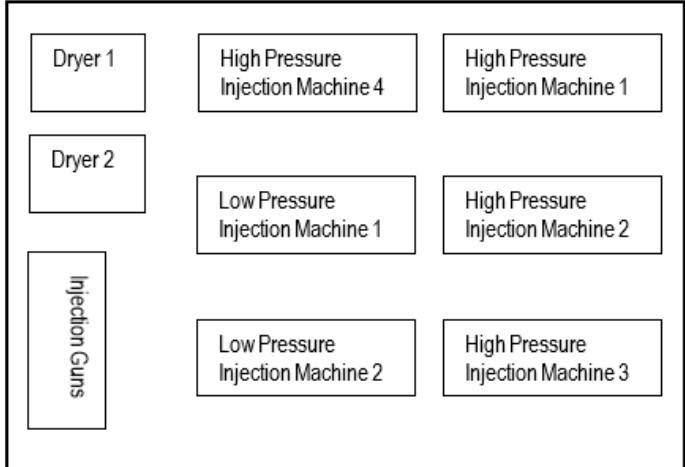

**Figure 1.** "L" and "O" layouts in the facilities.

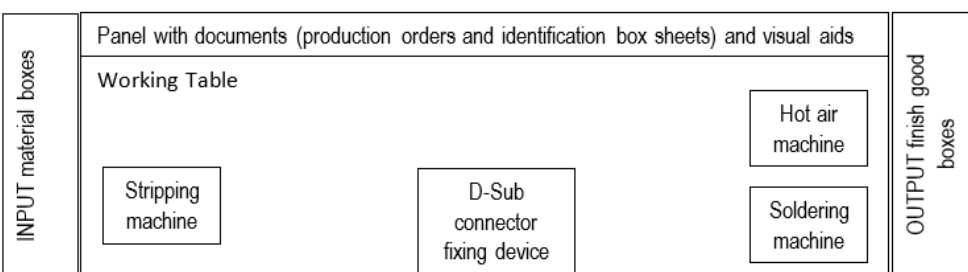

**Figure 2.** Job Shop layout for soldering manufacturing.

In the next chapter, it can be seen from the current process sequence and workflow diagram that the operator from the Job Shop is doing all the sequences; therefore, it will be difficult to balance the work or identify and limit of losses. It is clear that the VSM (Value Stream Mapping) cannot be applied now to this process. Of course, the Value Network Mapping [41] concept is supported to group the families and to merge the routings for

similar products based on the component family. However, the high mixing of products based on demand and high level of skills in soldering operations led to the decision of further improvement and the use of collaborative robots.

*Methodology*

For this study the DMADV methodology was used. In the first part of the paper, the Define phase of the methodology was used. Techniques such as Ishikawa, Voice of customer or SIPOC (Supplier, Input, Process, Output, Customer) were used to understand the process. In addition, they helped us to underline the needs of the client, the needs of the business and how the soldering process can be improved. After the use of Ishikawa technique, it was determined that the main causes of decreased productivity are: Lack of an intermediate check; Multiple suppliers for materials; Working Instructions defined only on paper; Lack of correct training from the beginning/wrong soldering method learned at the beginning; The workbench is not ergonomically designed; Withholding from the operator; Specialized training needed; Improper holding devices; Manual operated soldering station. Due to the data received from the company, it was decided that the soldering station will be automated.

In order to obtain the data, quantitative data have been used. They were used to define the correlation matrix between VOCs and functionalities and the scoring matrix. In measuring the data, all operators were involved and discussions with them led to the identification and solution of the problem. A series of improvements have been generated (Section 4.5. Designing the new production process) that will assure us that all the requirements have been taken into consideration.

The present methodology bases its results on the data collected from the company. It uses techniques, methods and tools to find the best solution regarding the possibility of automation the soldering process. The DMADV methodology uses the voice of the customer and voice of the business as selection criteria for answering the research question and the AHP prioritization model to redesign the process.

As a result, the robotic soldering cell, together with the preparation of the multi-wires cables, will ensure the improvement of the processes, not only in the capability and the quality of the soldering operation itself, but also in terms of the SMED methodology (easy selection of the pre-loaded programs and the dedicated RFID tags coded on the variant support of the cables and connectors; the OEE easily visualized and the issue is detected at early stage; the balancing of the lines can be achieved by analysis of the parametrization and the outputs, the one-piece flow will reduce the rework time lost; and the realization of production will fit with the schedule. Eventually, by regular VSM analysis, the inventory will be reduced and the WIP (work-in-process) leveling will decrease.

The problem this research is solving can be formulated as: Is it possible to automate a soldering process with 37 connectors having a high-mix, low-volume production characteristic? The difficulty in working with this type of product is the complexity of the process: first of all, the line must be adapted to produce a variety of products following the methodology of IPC/WHMA-A-620B standard. Secondly, for soldering, there are many complex pre-operations that need to be completed (see SIPOC diagram). Thirdly, the wires must be arranged into a specific order, taking into account the connector type and where it will be used. Each wire needs to be placed into a specific wire location according to a color code given for each wire type. Not all 37 wires are positioned on the same part: 18 are on one side, and 19 on the other side of the terminal, so it is necessary to flip the terminal after soldering the first row.

If this affirmation is true, the next question this research is trying to answer is: Is it possible to integrate a cobot within the line? The question appeared due to the characteristics of the process: high-mix, low-volume, and due to the complex pre-operations that need to be completed. In addition, if this process is normal for humans, when debating about the operations that can be performed by robots, the number and possibilities of managing a task are still limited.

## 3. Overview of the Workstation Planning

### 3.1. Collaborative Robots in Lines and Workstations

With the transformation of production lines into systems that facilitate as much individualized production as possible, it is necessary to implement the smart factory concept, which brings new approaches to the technological flow and the components that make them up, as shown in the figure below.

A requirement as shown in Figure 3, is the collaboration of the cobots in the lines and workstations with the human worker. Depending on the type of application (for example: assembly, soldering, screwing, etc.) and the division of work tasks between the human worker and the collaborative robot so that the process time must be optimal, there are several types of collaboration in the process lines:

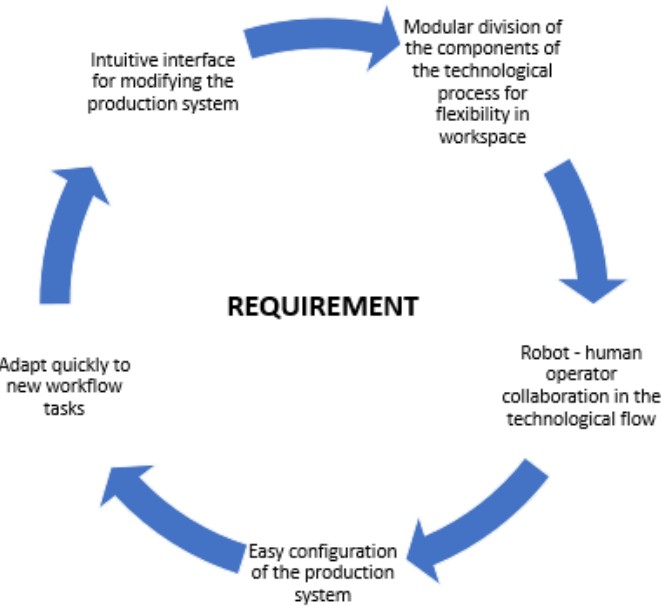

**Figure 3.** Requirements for building a smart flow production in factory.

Independent: Cobot and the operator share the same workspace, but their tasks are independent, the result being two different workpieces.

Simultaneous: The operator and the cobot perform two different operations on the same workpiece at the same time, thus improving the cycle time per piece and, implicitly, the productivity as well as the space saving.

Sequential: The collaborative robot and the worker sequentially perform operations on the workpiece, so that there is a dependency between them to complete the task. In the vast majority of cases, the cobot performs the first operations (e.g., hard work as pick and place, etc.), transferring the part to the operator to complete. This way of working has as the main purpose of improving the working conditions and the safety of the operator.

Supportive: The human operator and the cobot perform interactive operations on the workpiece at the same time. The degree of interaction is maximum, in the sense that the collaborative robot understands the requirements of the operator to provide help, considerably improving the ergonomics at work [42–44].

### 3.2. Workstation Planning

The high demand for products with a delivery time to the customer as short as possible and the quality conditions of excellent products results in an industrial factory to automate the process in the workflow. In this regard, the main purpose of industrial automation is to improve the production processes through: low and constant execution time, eliminating scrap from production and increasing the level of precision and quality of parts with high efficiency and high productivity. However, it must be considered that if automation

increases in the manufacturing process, the human work factor is reduced to a minimum. However, the current trend in the labor market regarding the number of people who want to work in an industrial plant, especially among young people, is constantly decreasing. Among the responsible factors are, on the one hand, the declining interest of the younger generation in professional trades, which results in a higher average age among employees in the industrial field, which has been growing in recent years, and, on the other hand, the demand for products in large quantities with low manufacturing costs. For this reason, the management strategy of the industrial sector, especially the automotive area, is to reduce the human factor in the process flow and the transition to automation and, implicitly, its advantages (as presented in previous chapters of this works), as well as the low level of qualification among the operators in such a way that the training period is short. In this case, the advantage of employers is the ease with which they can integrate a new employee in the production area. Another aspect that influences the production planning strategy is given by the activity sector. In the case of 2Connect Romania, it is represented by the electronic cables market. Representative for this sector type is the dynamic market, because many types of products must be manufactured in direct proportion to the number of products that have electrical and electronic components in their components. Therefore, the adaptation of the production is achieved at short intervals, depending on the new products that appear on the market, and most of the times the dedicated automated lines are not efficient, because they do not adapt so easily to the different types of products, and the manufacturing series on the piece is usually small or medium. Therefore, such an investment would not be amortized from a financial point of view (for this sector the manufacturing type, high-mix, low-volume, is representative). However, a solution is automatic flexible manufacturing cells, and the need to use collaborative robots is justified.

The challenge is to identify the operations in the technological flow where such robotic cells can be integrated in order to result in the maximum efficiency of the production with rapid amortization of the investment as well as the design of such an automatic cell, these being achieved by analyzing the operations in the section from an engineering point of view.

The cable is an assembly of components, such as isolated conductive wires, used to transmit electricity from one end of it to the other. Electric wire is a component of the cable that serves to conduct electricity from one point to another (from source to consumer). Depending on the requirements of the electronic application, the cables are of several types. 2Connect Romania manufactures classic coaxial, special, flat and hybrid cables, each with certain components in their construction.

For the cable production, 2Connect meets all the requirements and follows the methodology of IPC/WHMA-A-620B standard [45]. The standard provides the minimum requirements in terms of assembly, process control or inspection. The standard provides guidance regarding the minimum electrical clearance distance or the magnification aids, or the acceptance strand damage.

In Figure 4, the component parts of some of the types of cables manufactured in 2Connect are represented. It is observed that their common component is the wire (consisting of insulation and copper strands) which is an electrical component that has the role of conducting electric current from one point to another (from source to consumer). The wires are identified by the color/color combination of the insulation and according to the AWG (American Wire Gauge), which is a standard for measuring the diameter of the wire.

Depending on specific parameters (such as better resistance of wires, category of equipment, etc.), at the end of wires various electrical components called electrical terminals are mounted that have the role to make the connection between the current and consumer source.

In addition to establishing the electrical connection of the cables with the consumer, a good mechanical connection with it is mandatory. This means that the wire will be able to resist to a specified mechanical force according to each wire type or dimensions of the circuit/contact or terminal that must be resisted without causing damage to it. Due to the fact that the wire has 37 connectors, each one need to be intact without damage.

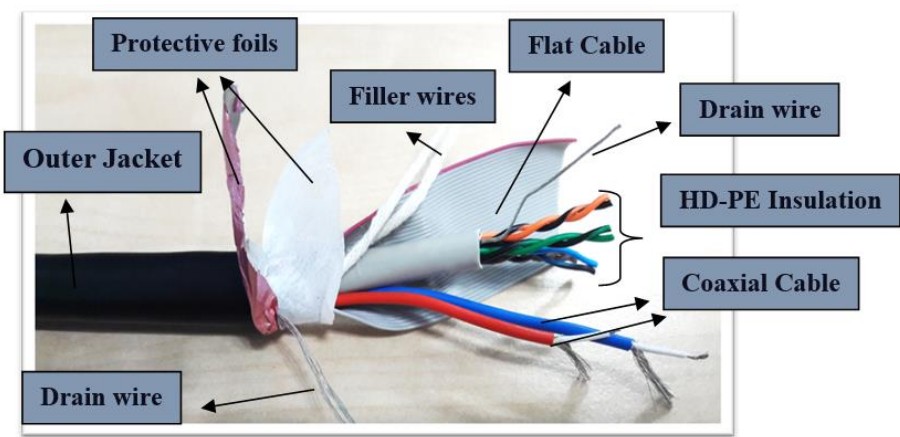

**Figure 4.** Example of hybrid electric cable made by 2Connect Romania.

## 4. The Lean Manufacturing Approach for the Development of A Robotic Soldering Cell

### 4.1. Soldering Design Process

For a better understanding of the process, the SIPOC diagram of the electronic cable manufacturing operations in the production plant was generated (Table 1).

**Table 1.** Suppliers, inputs, process, outputs, customers.

| Suppliers (5) | Inputs (4) | | Process (1) | Outputs (2) | | Customers (3) |
|---|---|---|---|---|---|---|
| | Description | Requirements | | Description | Requirements | |
| Cable Supplier En-Dev Dept Quality Dept Production Dept | Instruction Manual Client Drawing Dismantle tool | Continuity Risk | Dismantle of the cable | Dismantled cable | 37 mm ± 0.5 | Soldering Operator |
| En-Dev Dept Quality Dept Tool Supplier | Instruction Manual Client Drawing Cable Scissor | | Cutting of the wires | Wires at the correct length | 37 mm ± 0.5 | Soldering Operator |
| Stripping Machine Supplier En-Dev Dept | Instruction Manual Stripping Machine | | Automated Stripping of the wires | Stripped wires at the correct length | 2 mm ± 0.2 | Soldering Operator |
| Tin supplier En-Dev Dept | Instruction Manual IPC Tin bath Solder stripper | | Tin the wires into the solder pot | Wires are tined | 2 mm ± 0.2 | Soldering Operator |
| En-Dev Dept Connector Supplier Soldering wire supplier Quality Dept | Solder stripper IPC Instruction Manual Soldering Station Solder Wire | | Soldering of the wires | Wires connected to the pins' connector | Correct amount of tin Connectivity Colors coded according to Instruction manual | Soldering Operator |
| Thermo tubes' supplier En-Dev Dept | Instruction Manual Hot air Station Thermo Tube IPC | | Insulation of wires | Insulated wires | Insulation has to cover the stripped part of the wire | Soldering Operator |
| Soldering Operator | Insulated wires | | Visual check | Inspected insulated wires | Quality condition | Quality operator |
| Quality operator | Insulated wires | | Electrical Testing | Wires tested | Functioning conditions | Testing operator |

All necessary raw material (connectors, seals, etc.) are sent into production for the assembly stage. The first operation consists in stripping the cable and checking the integrity of the cable parts and orienting it in the working position. The length of the stripping is measured with the help of a specialized instrument. With the help of the cutter, the protective foil is removed and, depending on the type of product manufactured, the filling wires can be removed or not.

Depending on the specific customer requirements for each product, intermediate quality controls must be performed after each operation. In the case of stripping, the target condition is that the copper strands of the wire are not deformed or cut.

The terminals have different shapes and implicitly different methods of wire mounting. However, depending on certain similarities, they can be divided into groups (pins, connectors with built-in contacts, etc.). If you use the manual operation, the terminal is fixed in the cavity of the pliers. The other method consists in using the semiautomatic terminal mounting press. In both cases, the disadvantage is that if the terminals do not belong to the same groups, other pliers must be used, respective to the special press. Depending on the type of terminal, a visual inspection must be performed after the operation.

According to the SIPOC diagram (previously presented), the next operation is soldering, which is a technological process of bonding two or more metallic materials by melting and putting a solder (filler metal with a lower melting point than the others) into the joint. An important aspect to mention is the long training time of the operators in order to be able to achieve this at the quality standards imposed in 2Connect Romania. The wires on which the stripping was made, are put in a special degreasing solution after which they are introduced by the operator in the solder pot. Then, depending on the type of cable to be made, the soldering is performed, with the help of the soldering station, between the wires and the terminal contacts. At the end, with the help of the hot air station, the resulting joint areas are insulated with thermosetting tubes.

The operator must take into account certain conditions for performing a correct manual soldering operation, such as: the tip of the soldering device is chosen so that its surface is approximately equal to the wire or terminal to be solder; and the recommended temperature for the soldering tip is approximately 380 °C for 2Connect products. A low temperature can lead to incomplete soldering—cold soldering; a high temperature can damage the components to be soldered.

The quality of the process depends exclusively on the precision of the operator, this being a process with a high risk of injury during work, because it works with high temperatures, and the smoke released during the melting of the solder is toxic for a human operator that works 8 h/day.

As shown in the Figure 5, the solder must cover the strands of the wire (a), an insufficient amount of solder can short-circuit the cable (b). Another common problem (shown in Figure 6) can be the excess of solder, which leads to the melting of the terminal and the insulation of the wires, and results in the destruction of all cable components.

After insulating the soldered wires, the next operation consists into a series of phases that differ depending on the type of cable processed, such as: connecting electrical wires in connectors, mounting gaskets and inserting seals, mounting fuses and housings and the process of mounting ferrites.

After the operation of connecting the electrical wires has been completed, the operator must perform an intermediate quality control. To ensure a clear quality control protocol, the cable is divided into four areas (e.g., Figure 7a) to be checked. In the insertion area 1, they must check if the wires are inserted up to the end of the connector, in the wire end check area 2, if the wires are cut evenly is checked, the insulation fixation zone (area 3) is observed if the insulation is inserted until after contact and the insulation crimping area (area 4). Figure 7b shows frequent errors that are discovered during the intermediate technical control, such as: cut wires, pinched insulation, the wires are not inserted to the end of the connector, and the crimping of the insulation is not tight enough.

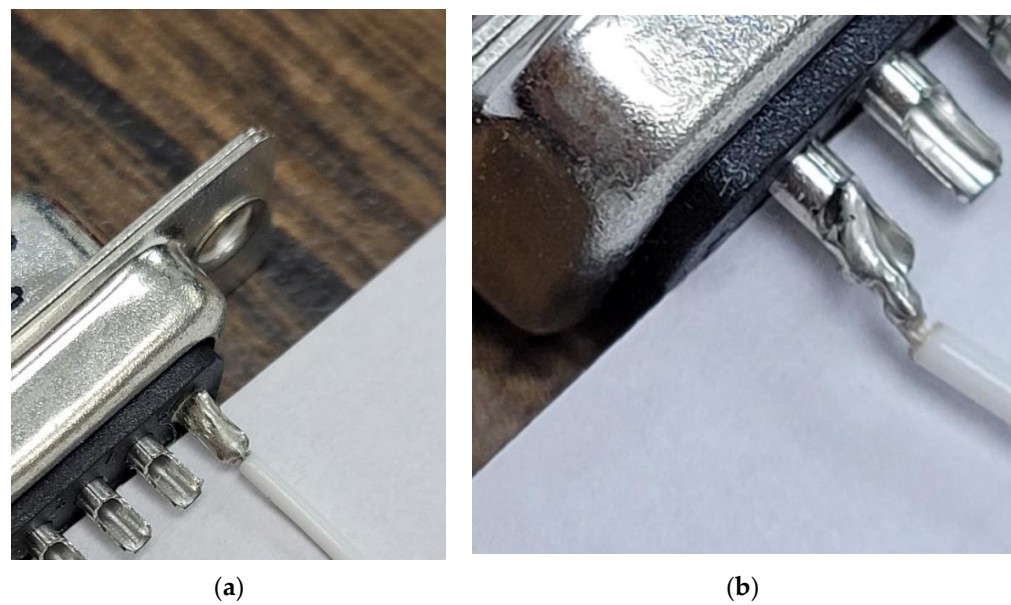

**Figure 5.** Example soldering: (**a**) target condition; (**b**) unacceptable condition.

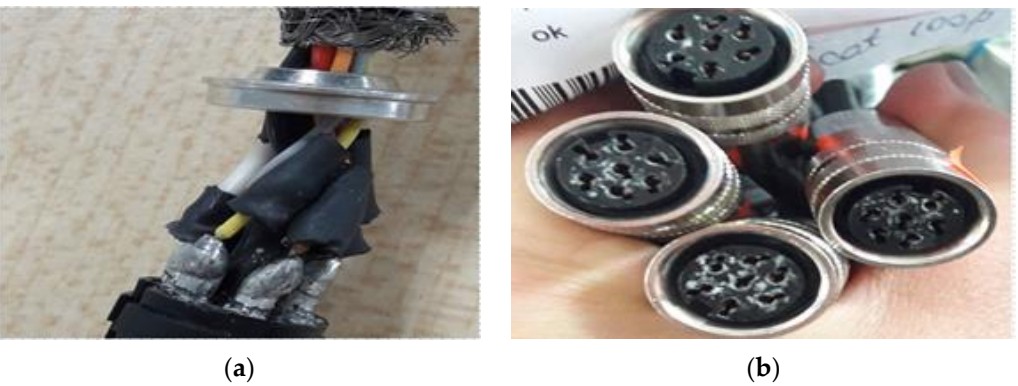

**Figure 6.** Example of scrap parts in 2Connect Romania: (**a**) Wire insulation problems; (**b**) Melted terminal.

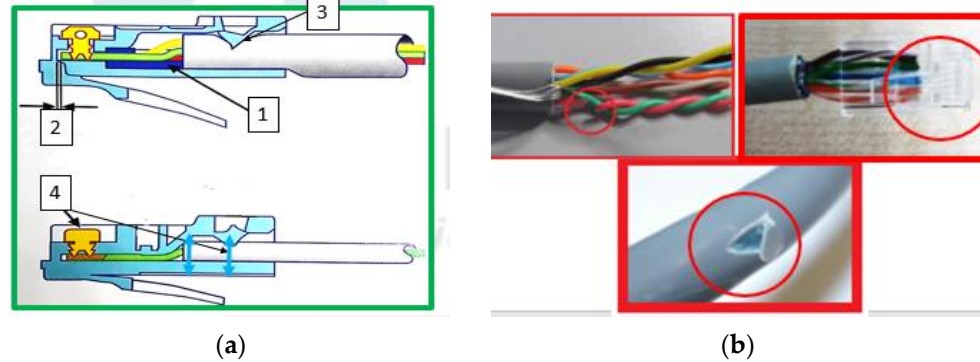

**Figure 7.** Example connecting: (**a**) target condition; (**b**) unacceptable condition.

After preliminary checks, the electrical controller is made to see the continuity of the transmission of electric current from one end of the cable to the other. The electrical check of the cable is carried out at a tester by introducing all the connectors in the counterpart and connecting the contacts to some pliers. The counter pieces contain some pins that are in contact with the terminals/contacts to show their continuity. The working process is: take



the cable from the basket, scan the reference on the crate sheet to open the electronic work manual, visually check and connect to the tester counterparts and perform the electrical test. If the cable is satisfactory, has continuity and does not short circuit, the message "PASS" (also green) appears on the screen. If the cable does not have continuity or makes a short circuit, the message "FAIL" appears (for other types of testers, the red LED lights up), then a retest is made, checking not to touch the counterparts. If the same message appears, the cable is identified by the red label and placed in the red box.

### 4.2. Identification Selection of Workstations for Cobot Integration

From the description of the production flow presented in Section 4.1, it can be seen that the operation steps are different depending on the type of cable to be manufactured. Given that the production is of the high-mix, low-volume type, i.e., a large number of cable types are manufactured in small quantities at irregular intervals, it is particularly difficult to establish where the key points of process optimization are that have positive effects on all parts and are manufactured without the help of a complex multi-criteria analysis.

The technique of advanced multicriteria analysis has, as a major advantage, the objective character of its results because: it is taken into account, through a simple mathematical scheme, that the relative position of two criteria can have only three main situations: one criterion is more important than the other, one criterion is as important as the other and one criterion is less important than the other; the analysis being performed separately for each criterion [46].

For the chosen theme, the Qualica software package was used to be efficient in data processing; the project being a complex one, it is part of a more detailed analysis. In the following, some relevant steps for the approached topic will be presented.

The first step of the DMADV method is to define the analysis criteria of the entire process that the company aims to redesign by establishing the customer demands and the importance of each one for possible optimization in the future.

In the Table 2, there are eleven criteria to be analyzed, obtained after brainstorming sessions with the team.

**Table 2.** Voice of Customer. (The customer is represented by the management).

| VOC ("Voice of Customer") |
| :---: |
| Decreasing the cycle time/operation |
| Decreasing the number of scrap/operation |
| Increasing the process accuracy |
| Increasing operator's safety |
| Decreasing the number of human operator training hours/operations |
| The degree of automation/operation |
| The impact of automation upon the process flow |
| Decrease bottlenecks in the system |
| The complexity of re-designing the cell-workstation |
| Increasing the productivity/operation |
| The complexity development of the robotic cell |

The following image highlights the AHP ("Analytic Hierarchy Process") analysis for the VOCs already mentioned. Depending on the input criterion, a mark from 1 to 9 will be assigned. If both input and output requirements are equally important, then the value assigned is 1. The results are illustrated in Figure 8.

Figure 8 shows the results of the analysis, which consists in ranking and sorting each criterion according to its importance upon the process to be redesigned.

The requirement with the most importance is "increasing the productivity/operation", having a percentage of 15.4, which justifies the need to redesign the entire process. The first requirement is followed by "decreasing the number of scrap per operation", with an importance of 12%, which is directly related to it, because in the company, the degree of error produced by human factor involved in the process is extremely high. In addition,

the third important item from VOCs is "decreasing the number of human operator training hours/operation" with a percentage of 11.2, which indicates that the human factor is to be considered when redesigning the process, because nowadays all companies in the cable manufacturing field are facing a crisis regarding qualified workers for the production sector.

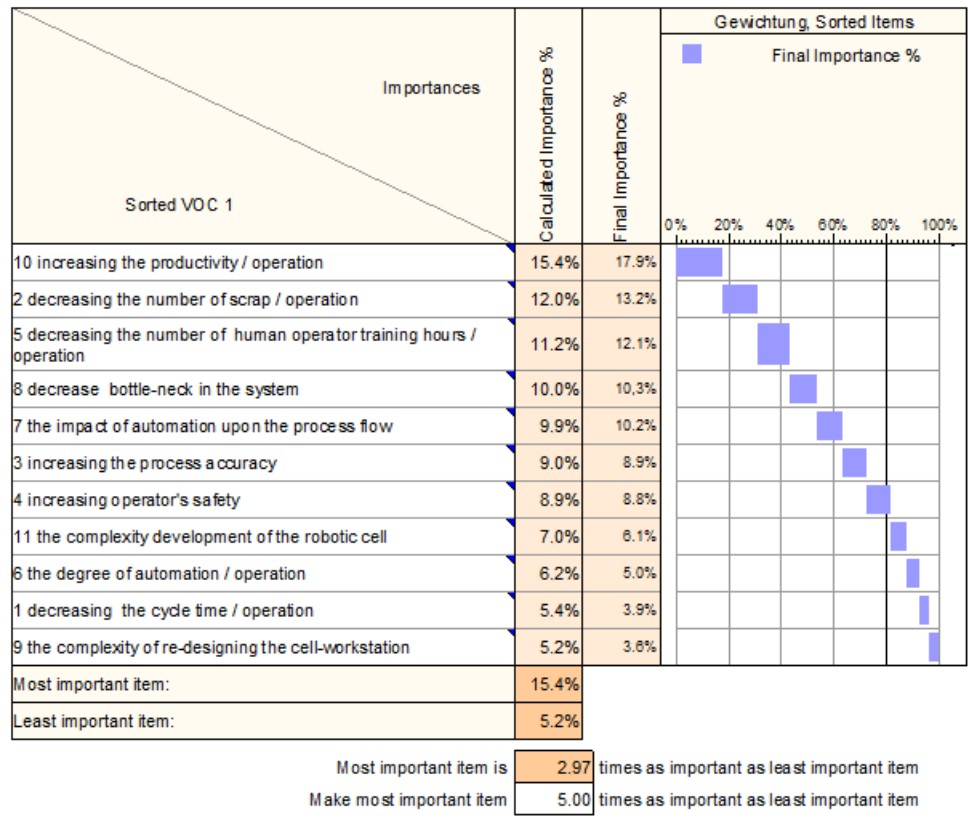

**Figure 8.** The sorted results of the Analytic Hierarchy Process Analysis.

On the other hand, the least important requirement is "the complexity of re-designing the cell workstation", with a percentage of only 5.2, implying that the company is not concerned with the difficulties of improving the process, but rather with the benefits associated with its implementation. On the penultimate place is situated "decreasing the cycle time/operation", with an importance of only 5.4%, because the manufacturing in this company functions on the "high-mix, low-volume" topology, which means that they have a high variety of manufactured products, but with a low production capacity, and reducing the cycle time without increasing the product quality is not significant.

### 4.3. Automation Requirements and Current Challenges

The next step is to establish all specifications based on the customer's definition of what they require. The specification describes the product or service in quantifiable terms, enabling the collection and comparison of data against the defined requirements. This is crucial for achieving that the final product fulfils the already mentioned customer demands from the first step or the VOCs.

After ranking the VOCs, 15 performance requirements have been identified:

- The first performance requirement outlines the fact that the effective time of each operation should decrease to an ideal value of 75% of the cycle time, but it is accepted the decrease down to 90%.
- The second refers to the fact that the scrap should decrease to 60%, the ideal value being around 30%, this leading to an error rate withing the quality standards, according to the company management.
- The third quality requirement is that the soldering accuracy should increase up to an ideal value of 60%, with a marginal value of 30%, this being given by the fact that the human operator error rate would significantly decrease once the collaborative robot is integrated in the workstation.
- The fourth one refers to the safety of the operator that should be increased up to a minimum of 75% once the robotic system is integrated. The ideal value for this requirement can be only 90%, because even if the collaborative robot is improving the working conditions for the human operator, the perfect value will never be reached for the soldering process.
- The fifth quality requirement is the number of training hours, which should decrease to 15% of the training duration for the manual soldering. The ideal value is 10%.
- The sixth outlines the fact that the entire process should involve more than one automated devices, but the ideal value is five automated devices.
- The seventh quality requirement refers to the effective time needed for the manufacturing of a single part, which should decrease under the value of 90%, and the ideal value is 75%.
- The eighth highlights the fact that the bottlenecks in the system should decrease more than 14,400, but the ideal value should be 7200 s for reaching the optimal productivity, according to the company management.
- The ninth quality requirement is the reconfiguration of the workstation, which should take place in less than 30 min, but the ideal time would be 10 min.
- The tenth quality requirement refers to the fact that the soldering productivity should reach an ideal value of 70%, yet the minimum threshold to be reached with the integration of the collaborative robot must not be less than 30%.
- The eleventh outlines that the automated equipment should be able to make software changes in less than ten minutes, the ideal value that should be reached being 5 min.
- The twelfth quality requirement refers to the fact that the accuracy of the manipulation should increase up to the ideal value of 60%, the minimum value for the manipulation using a collaborative robot being 35%.
- The thirteenth outlines the fact that in the entire process should have more than one safety system, but the ideal value is five.
- The fourteenth quality requirement refers to the fact that in the entire process should have more than one safety system, but the ideal value is five.
- The last one is that the mean time between failures should be greater than 2500 h, with an ideal value of 4800 h.

A correlation between the customer needs or VOCs, and the CTQs or the quality requirements, which are rated, is presented in Figure 9. After completing the entire matrix, following the analysis, the product capabilities were reached, which refers to the importance of each customer need in correlation with the company quality requirements for the products.

After analyzing the process, a brainstorming meeting was initiated. At the meeting, operators, the production manager, the quality manager and two other stakeholders participated. The purpose of the meeting was to determine the possibilities to increase the production. In this direction, an Ishikawa diagram was generated (Figure 10).

**QFD1**

- ✍ 9.00 strong correlation
- ✊ 3.00 some correlation
- ☝ 1.00 possible correlation
- ☺ 0.00 (possible negative correlation)

Column headers:
1. cycle time less than 90%
2. number of scrap less than 60%
3. soldering accuracy more than 35%
4. operator's safety more than 75%
5. number of training hours less than 15%
6. more than 1 automated device
7. cycle time/part less than 90%
8. bottle-neck decreased with at least 4h
9. work-station reconfiguration less than 30min.
10. the soldering productivity more than 30%
11. automated equipment software changed in less than 10min.
12. manipulation accuracy more than 35%
13. at least one safety system
14. at least one emergecy system
15. mean time between failures greater than 2500 h

| Rows | Number of significant relationships |
|---|---|
| 1 decreasing the cycle time / operation | 6 |
| 2 decreasing the number of scrap / operation | 6 |
| 3 increasing the process accuracy | 8 |
| 4 increasing operator's safety | 4 |
| 5 decreasing the number of human operator training hours / operation | 6 |
| 6 the degree of automation / operation | 14 |
| 7 the impact of automation upon the process flow | 10 |
| 8 decrease bottle-neck in the system | 5 |
| 9 the complexity of re-designing the cell-workstation | 4 |
| 10 increasing the productivity / operation | 9 |
| 11 the complexity development of the robot... | 6 |
| **Significant relations** (columns 1–15) | 6, 6, 5, 2, 7, 10, 6, 5, 5, 6, 5, 4, 4, 4, 3 |

**Figure 9.** Matrix of Quality Function Deployment (QFD) methodology.

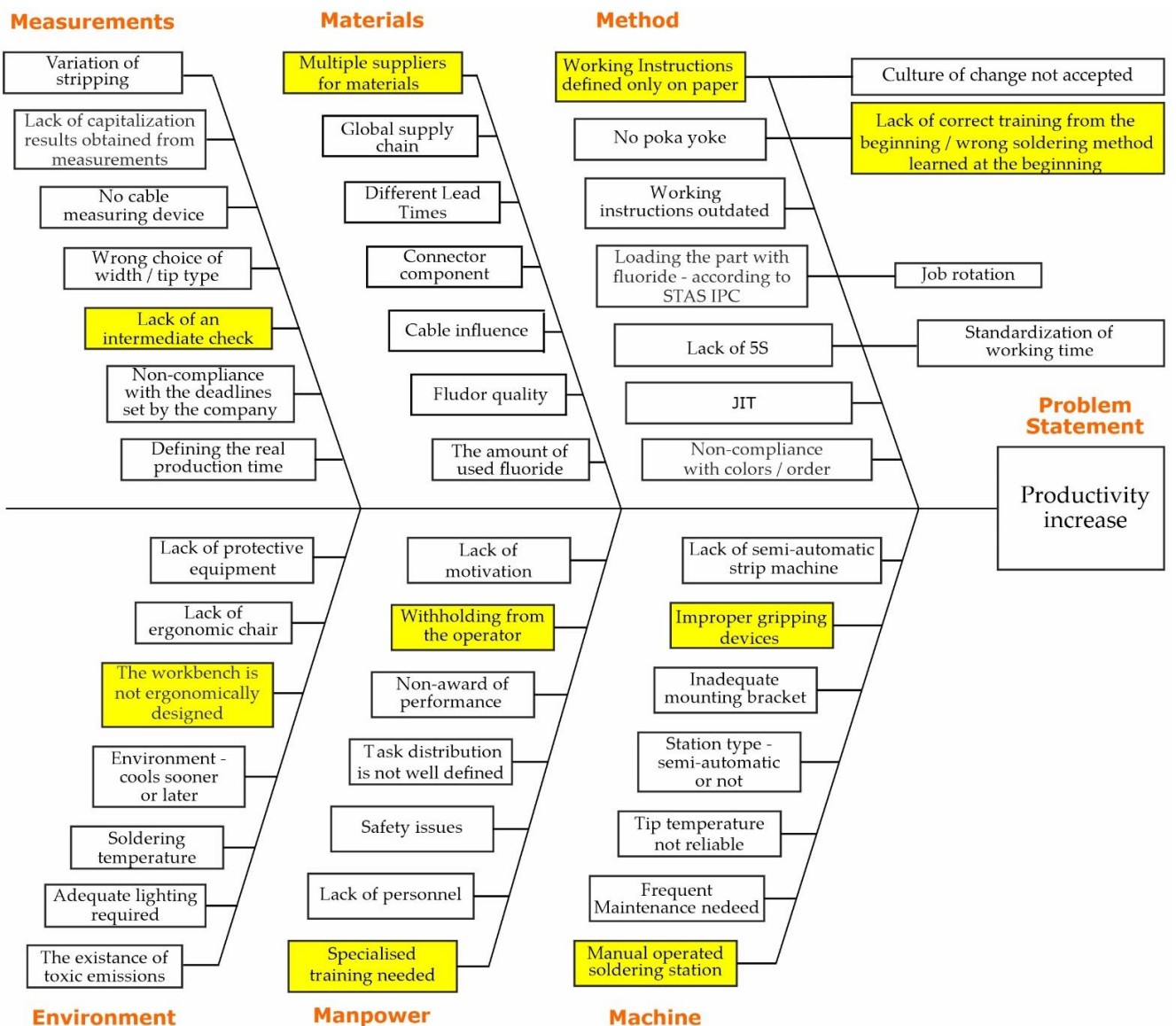

**Figure 10.** Ishikawa diagram.

As a result of the analysis, it was determined that the main causes of decreased productivity are:

- Lack of an intermediate check. If an intermediate check was introduced, the number of scraped parts will be reduced. Due to the fact that the process has only one visual check, at the end of it, mistakes may appear during the cutting, dismantle or stripping the wire.
- Multiple suppliers for materials. The variation of wires/housings/tubes differs depending on the suppliers. Working instructions must be defined for each supplier, depending on the specifics of the elements that differ.
- Working instructions defined only on paper. Each operator has a level of experience and tend to understand the process needs better than the standard. For different operations, they tend to diminish their importance, in this way non-compliant products can be obtained.
- Lack of correct training from the beginning/wrong soldering method learned at the beginning. When an operator is used to a certain method that can be incorrect, it is difficult to teach and become accustomed to the correct one. Usually, the problem appears for operators that were trained wrong at the beginning of their career and were not corrected during years.

- The workbench is not ergonomically designed. The workbench is full of devices that are used along the process, creating a clutter space. An organized workbenches will help the operator to produce more parts, of better quality, without having problems finding the devices.
- Withholding from the operator. The attitude toward change is not accepted and embraced.
- Specialized training needed. Regular trainings, with examples and practical execution will help the operators to see where they make mistakes.
- Improper holding devices. In this moment, there is not any custom holding device designed for assembling the cable. The operators are helped by a vice, but it does not meet the requirements of the operators, who largely have to improvise to complete the assembly. In this way, the time spent assembling the cable is very long.
- Manual operated soldering station. The assembly of the part is performed exclusively manual. The operators are the ones that fix the terminal in the cavity of the pliers. Due to this fact, there are many scrapes and the time spent with assembly is high.

### 4.4. Data Collection and Data Analysis

For a better understanding of the process or product, a list of areas that require modification or improvement was created, which includes both the product and the manufacturing process. At this point, a design may be developed and assessed. In this stage, the goal is to come up with alternatives, called concepts, evaluate those alternatives and implement those options that would improve the process or the eventual outcome.

In the Figure 11, the functional input data or the functionalities that the system should be able to achieve with the integration of a collaborative robot and the redesigning of the process are presented. The 12 functionalities were chosen according to the Lean Manufacturing methodology and the requirements listed by the 2Connect Romania team, that aim for eliminating any kind of waste within the production level of the company. Therefore, the improvement upon the process will bring higher value for the customers while reducing waste.

| What needs to be done? << - >> How can it be done? |
| --- |
| By reducing training time |
| By descreasing manipulation time |
| By decreasing assembly time |
| By decreasing stripping time |
| By decreasing cutting time |
| By decreasing dipping time |
| By decreasing soldering time |
| By decreasing crimping time |
| By decreasing insulation time |
| By decreasing overmolding time |
| By decreasing scrap |
| By decreasing number of deffective parts |

**Figure 11.** The input for the Functional Analysis.

However, along with the functions in redesigning the process, there are a series of constraints highlighted in the Figure 12 that will guide us to reach the reach the optimal solution.

In Figure 13, the correlation matrix between the customer needs or VOCs identified in the first subchapter and the functionalities that are the input data in this stage of the DMADV method can be observed. It was completed using the Quality Function Deployment (QFD) methodology, the ranking system being a collection of icons indicating whether a correlation is strong, moderate, or weak.

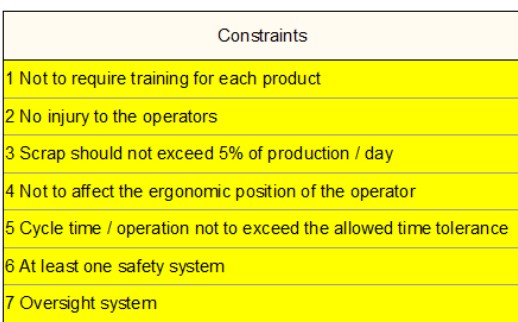

**Figure 12.** The constraints.

Following this analysis, the importance of each customer need was determined and examined in correlation with the functionalities of the redesigned process.

In the same manner and part of the same methodology, Figure 14 highlights the correlation matrix of the quality requirements or CTQs and the functionalities determined in the current stage of the DMADV method used for redesigning the current process. For each CTQ, the importance is determined in percentages according to the number of significant relations that exists between it and the functionalities that it correlates with.

After completing the above matrix, the most important six functionalities are the following:

- By decreasing soldering time —14%
- By decreasing stripping time—14%
- By decreasing assembly time—13%
- By decreasing overmolding time—13%
- By decreasing crimping time—12%
- By decreasing training time—12%

After performing the concept proposal analysis, the four concepts (manual cutting, manual stripping, manual soldering and manual connecting the wires) have been eliminated from the calculation of the new production flow design for 2Connect Romania (Figure 15).

The first concept proposed will consider the base concept used for further comparations between the three of them, in order to choose the optimal one. The base concept is called "1.1.2 + 2.2.2 + 3.3.2 + 4.4.2 + 5.5.2 + 6.6.2 + 7.7.1" and it consist of the following concept fragments: cutting with automated device, automated striping, crimping with automated device, soldering with automated device, insulation with automated device, connecting with automated device (mechanical/pneumatic) and over molding with automated device.

The name of the second concept is "1.1.2 + 2.2.2 + 3.3.1 + 4.4.2 + 5.5.1 + 6.6.2 + 7.7.2 1" and it consists of: cutting with automated device, automated striping, manual crimping, soldering with automated device, manual insulation, connecting with automated device (mechanical/pneumatic) and over molding with automated device.

The third concept is called "1.1.3 + 2.2.3 + 3.3.3 + 4.4.3 + 5.5.3 + 6.6.3 + 7.7.2" and it consist of: cutting with robotic cell, striping with robotic cell, manual crimping, soldering with robotic cell, insulation with robotic cell, connecting with robotic cell and over molding with robotic cell.

In the two images, Figures 16 and 17, there are the concept scoring matrices used in the analysis. When it comes to design and models, a concept scoring matrix is far more beneficial in identifying which choice is the most efficient. A weight percentage is assigned to each of the criteria based on the importance of the factor in question. According to the criteria that have been established, the design with the highest total is the best option.

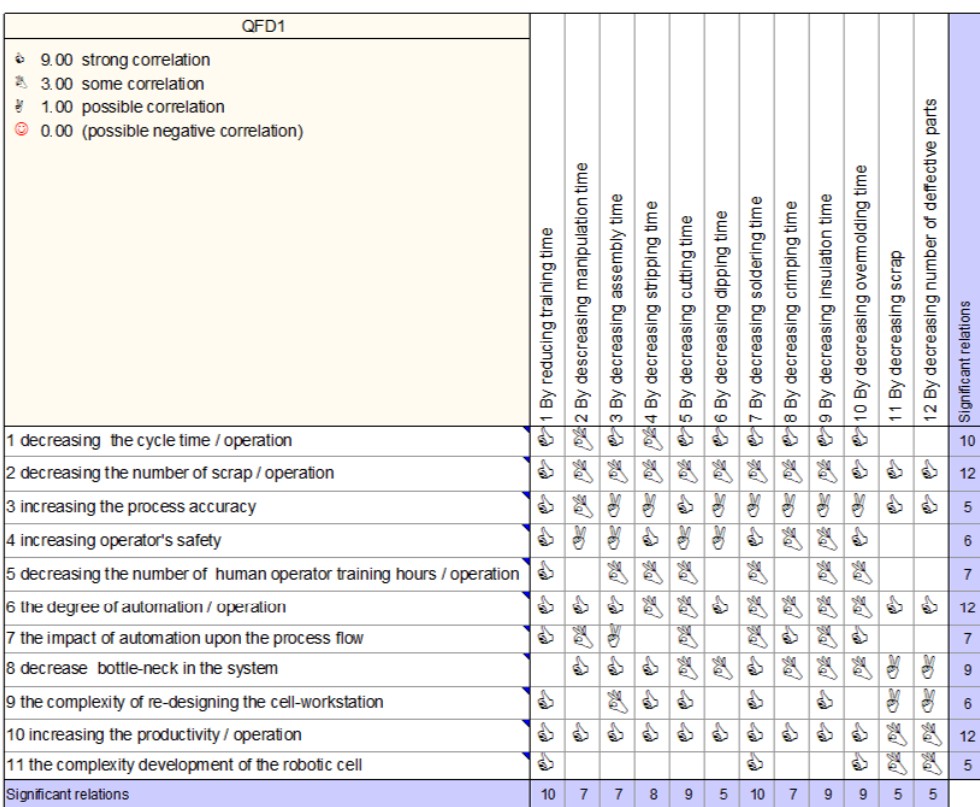

**Figure 13.** Correlation matrix between VOCs and functionalities.

**Figure 14.** Correlation matrix between CTQs and functionalities.

| Concept Fragments | 1.1.2+2.2.2+3.3.2+4.4.2+5.5... | 1.1.2+2.2.2+3.3.1+4.4.2+5.5... | 1.1.3+2.2.3+3.3.3+4.4.3+5.5... | Status | |
|---|---|---|---|---|---|
| 1.1 Manual cutting | false | false | false | | |
| 1.2 Cutting with automated device | true | true | false | | |
| 1.3 Cutting with robotic cell | false | false | true | | |
| 2.1 Manual striping | false | false | false | | |
| 2.2 Automated striping | true | true | false | | |
| 2.3 Striping with robotic cell | false | false | true | | |
| 3.1 Manual crimping | false | true | true | | |
| 3.2 Crimping with automated device | true | false | false | | |
| 3.3 Crimping with robotic cell | false | false | true | | |
| 4.1 Manual soldering | false | false | false | | |
| 4.2 Soldering with automated device | true | true | false | | |
| 4.3 Soldering with robotic cell | false | false | true | | |
| 5.1 Manual Insulation | false | true | false | | |
| 5.2 Insulation with automated device | true | false | false | | |
| 5.3 Insulation with robotic cell | false | false | true | | |
| 6.1 Manual connecting the wires | false | false | false | | |
| 6.2 Connecting with automated device (mechanical/ pneumatic) | true | true | false | | |
| 6.3 Connecting with robotic cell | false | false | true | | |
| 7.1 Overmolding with automated device | true | true | false | | |
| 7.2 Overmolding with robotic cell | false | false | true | | |

**Figure 15.** Morphological Box.

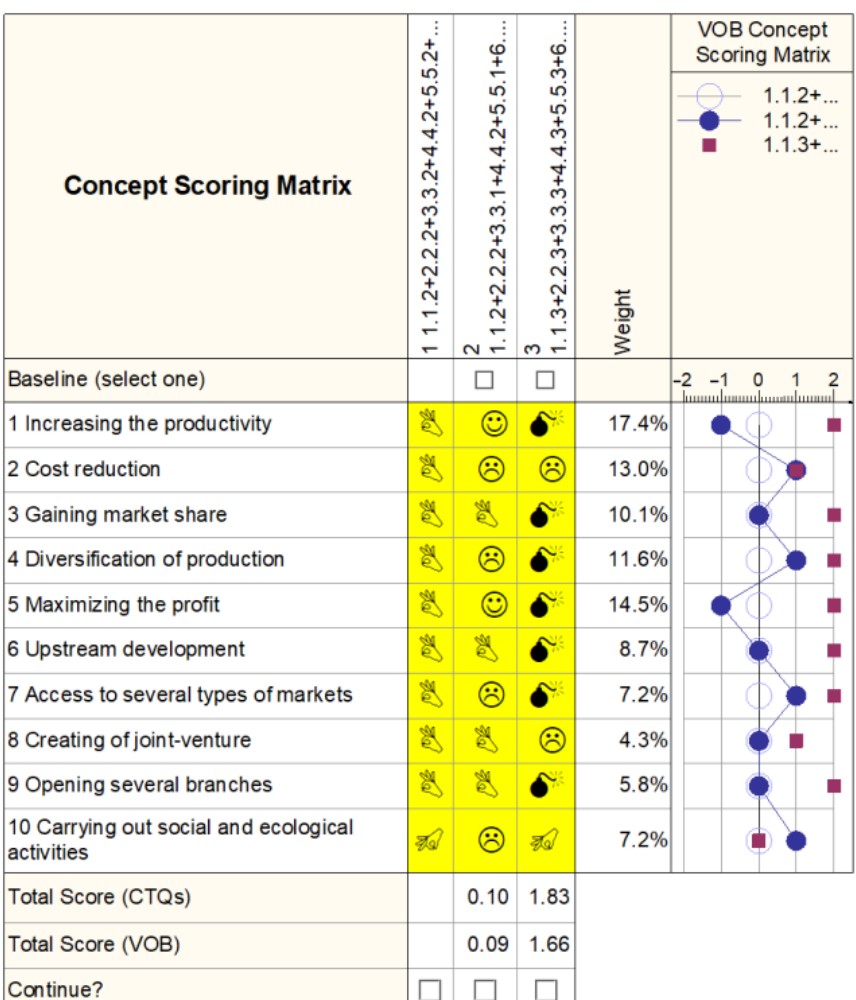

**Figure 16.** Concept Scoring Matrix—VOB.

For both concept scoring matrices, the last two variants of concepts for redesigning the entire process were compared with the first, which represents the baseline.

The outcome of the analysis showed that the third concept is the right approach in terms of reaching the performance aspects provided by CTQs, with a total score of 1.83, along with getting a great match, with a score of 1.66, with the limitations and management strategy of the organization.

In light of this analysis, together with the company's management, it was determined that robotization would provide the greatest benefits for the soldering operation. This is justified because the soldering operation generates the greatest losses for the company due to the fact that it is a complex process, requiring extensive training for operators, with long operation times, and the highest number of complaints from customers dealing with defects that were mostly experienced during this process.

*4.5. Designing the New Production Process*

Using the information gathered during the analysis step, this phase of the DMADV concentrates on obtaining the best option attainable to identify the fundamental causes of the issue. Before the solution is deployed in the reality, it is optimized and any potential errors are resolved. With each adjustment, the analysis phase is repeated to ensure that the new design meets the requirements for each of the given features.

The optimization of the concept, or more accurately, the re-optimization of the third concept from the previous chapter, is the main subject of this phase. The components used for the soldering process to be redesigned are presented in the Figure 18, along with

information such as quantity, unit, specification, price per unit and the supplier for each component used.

In the Figure 19, the correlation matrix between the design elements and the functions that are the input data in the previous stage of the DMADV method can be observed. It was completed using the Quality Function Deployment (QFD) methodology, the ranking system being a collection of icons, indicating whether a correlation is strong, moderate, or weak.

Following this analysis, the importance of each component used for redesigning the process was determined and examined in correlation with the chosen functions.

After establishing the importance of the criteria in the previous subchapter, the information from the process flow according to the identified needs and requirements will be filtered in this part, in order to identify the area of production where robotization would bring maximum benefits.

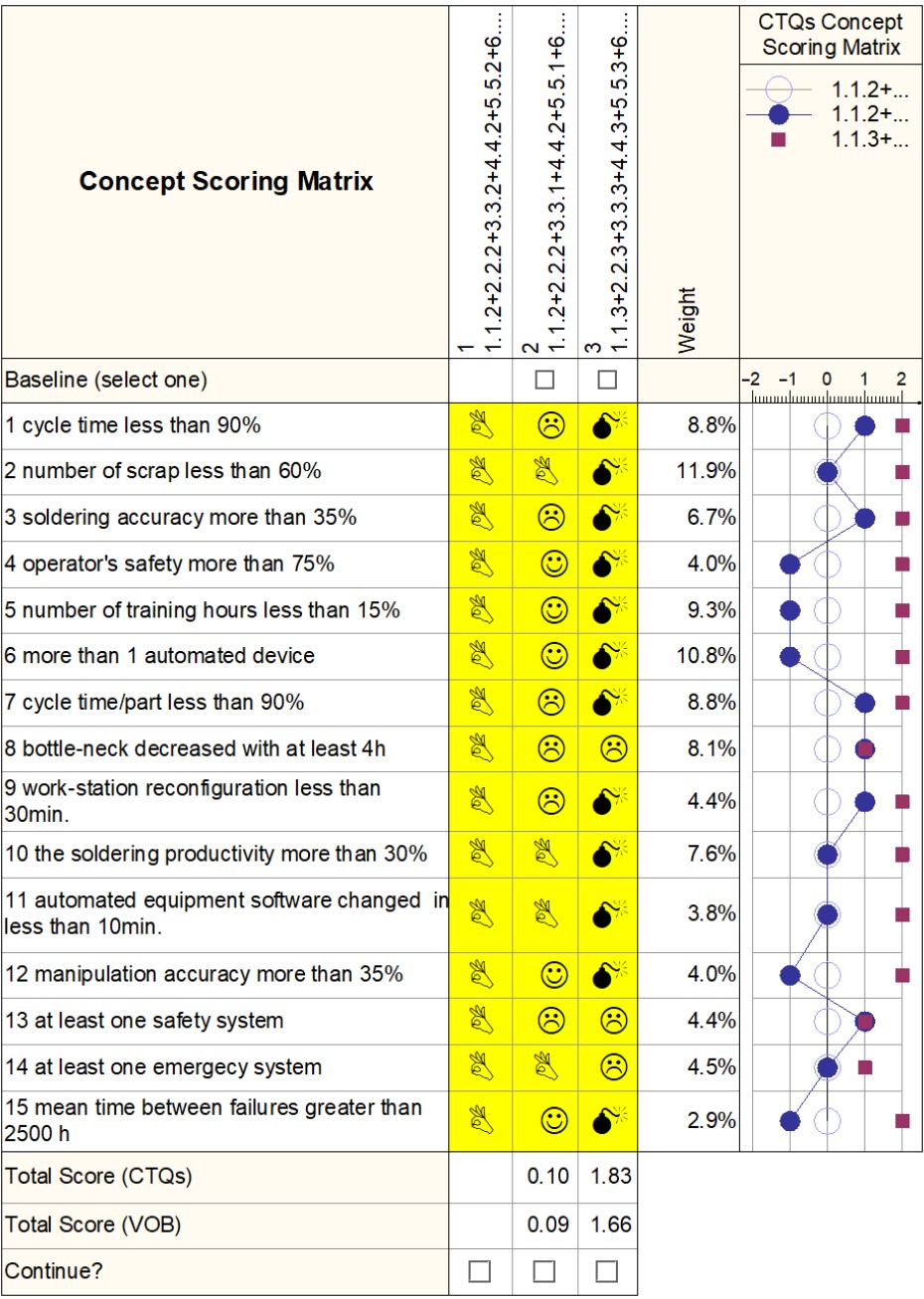

**Figure 17.** Concept Scoring Matrix—CTQ.

| Design Elements | Qty | Unit | Specification | Supplier | Price per Unit | Manufacturing Costs | Material Direct Costs | Overhead Costs % | Total Costs |
|---|---|---|---|---|---|---|---|---|---|
| 1 Collaborative robot arm U R5e | 1.0 | buc | Payload 5 kg | OVISO ROBOT... | 23,500.00 | 600.00 | 23,500.00 | 0.00 | 24,100.00 |
| 2 Clamping device | 1.0 | buc | reconfigurabe | | 100.00 | 50.00 | 100.00 | | 150.00 |
| 3 Gripping device | 1.0 | buc | 2 fingers | | 75.00 | 50.00 | 75.00 | | 125.00 |
| 4 Safety system | 1.0 | buc | | OVISO ROBOT... | 3 000.00 | 30.00 | 3 000.00 | | 3 030.00 |
| 5 3D vision camera | 1.0 | buc | Color recognition | OVISO ROBOT... | 3 500.00 | 35.00 | 3 500.00 | | 3 535.00 |
| 6 Soldering kit station | 1.0 | buc | WXR3 | OVISO ROBOT... | 2 773.00 | 27.00 | 2 773.00 | | 2 800.00 |
| 7 Stripping device | 1.0 | buc | tme.com | | 250.00 | 25.00 | 250.00 | | 275.00 |
| 8 Solder | 1.0 | buc | tme.com | | 0.50 | 0.04 | 0.50 | | 0.54 |
| Total Cost: | | | | | Total Cost | 817.05 | 33,198.50 | 0.00 | 34,015.55 |

**Figure 18.** Final Concept Components—Last optimization.

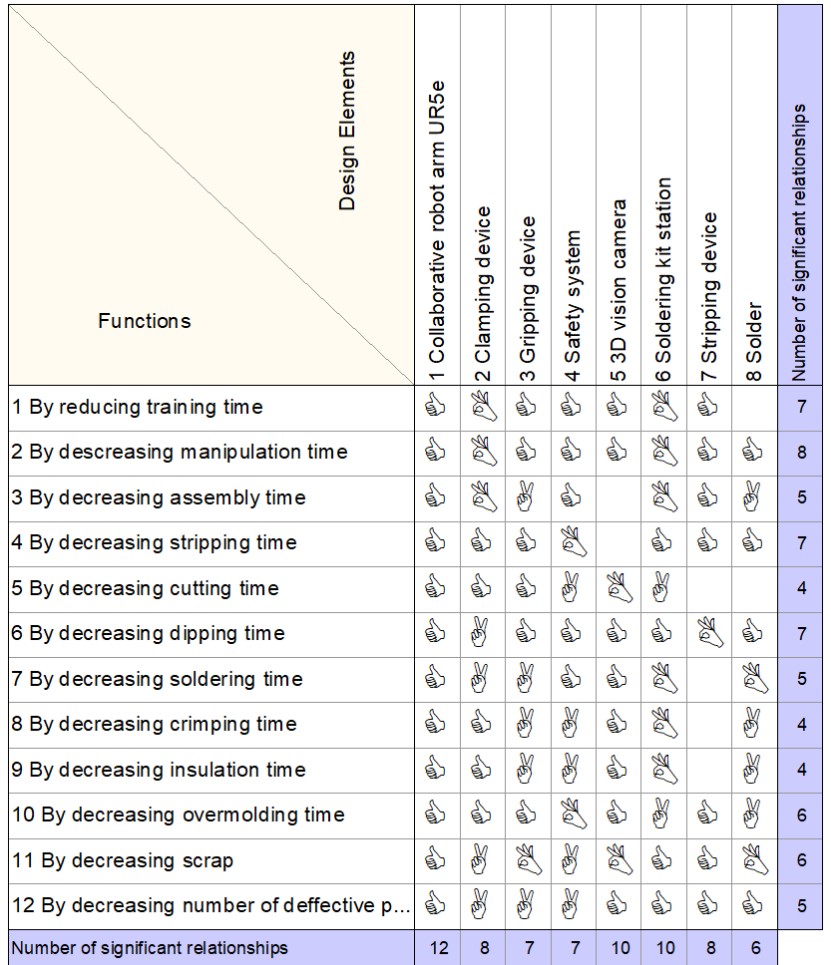

**Figure 19.** Correlation matrix between Design Elements and Functionalities.

Figure 20b shows the functions that the collaborative robot must perform, these being established based on the requirements (in this case the phases of the work operations performed by a human operator in 2Connect Romania). The constraints that may occur due to the functions identified must be taken into account; all this information introduced

in Qualica will help create several constructive variants of the re-design of the production process and, implicitly, to establish the robotization zone/zones.

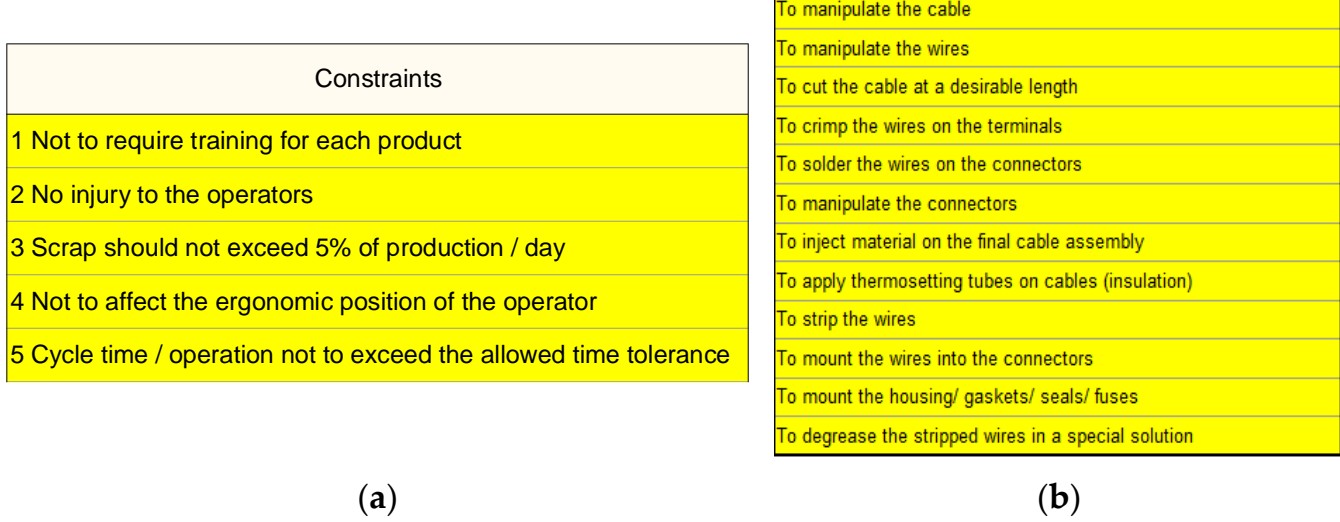

**(a)**                      **(b)**

**Figure 20.** Necessary collaborative robot constraints and functions: (**a**) constraints; (**b**) functions.

The next step is to establish the requirements from the company's management (Voice of Business), as well as the expectations they have regarding the robot implementation project. These requirements are also classified according to importance. Among the most important are: maximizing the profit, cost reduction, gaining market share, upstream development, etc. After this step, each operation in the technological flow will be divided into three possible blocks (operation performed manually, with automated device or robotic cell), and they will go through the filtering from VOC and the functions and constraints noted in Figure 20.

An example in this sense is given in Figure 21, where the way in which the cable manufacturing operations could be achieved is analyzed to see if it respects the constraints. Thus, the variants that are marked with a red dot do not meet the constraints, so they will be excluded from the constructive variants of re-designing the layout, the variants marked with a green dot are those that fully meet the constraints, so they are considered optimal and those marked with a yellow dot require a deeper investigation, so they will also be taken into account.

The three concepts rejected in the calculation of the new design of the manufacturing flow from 2Connect Romania are also confirmed by the Quality department, because most of the scraps result from these operations. For example, due to the fact that the insulation of the cables is made of a thick plastic material, the manual cutting can be achieved perpendicular to the cutting scissors, this being accentuated by its copper strands. For this reason, the operation must be repeated in such a way that the material is wasted, with the possibility of injury to the operator. Another factor that emerges from the analysis presented above is manual soldering. This process requires extensive training time for the operator to meet the standards required by the customer for each cable (approx. 2–3 months). The ergonomic position of the operator is affected, and other effects are also the cause of danger of accidents (working with solder at 350–420 °C) as well as the smoke released during the process. Another aspect to mention is that this operation has a long cycle time compared to other operations in the flow, and also many scrapes occur due to its complexity (wire damage, short circuit, etc.). The last concept eliminated is manually connecting the wires, due to the fact that there are a variety of parts manufactured in the company, and the errors

that occur frequently are also due to the reversal of the pins or the wrong connection which leads to short circuit.

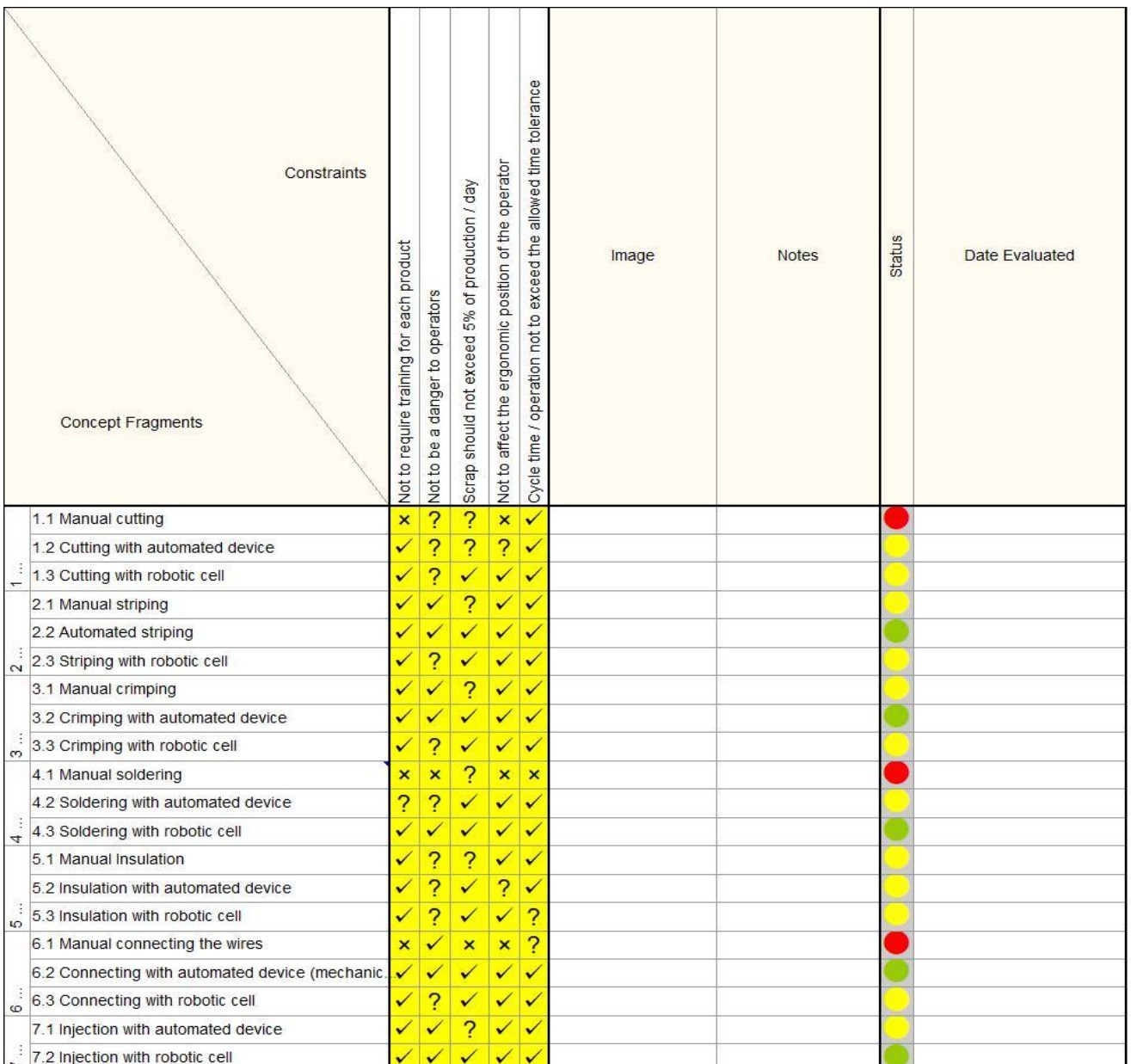

**Figure 21.** Analysis of constructions in relation to constraints.

To conclude from Figure 21 is that a first direction is the optimal result (green dot) for the soldering and/or injection operation with a robotic cell. The other optimal variants are already applied in 2Connect Romania at the time of the analysis in Qualica (striping, crimping and connecting with automated device). In order to see which is the optimal variant of optimizing the cable manufacturing system, based on the concepts from Figure 21 (green and yellow dot), three constructive variants of production flow were proposed as follows:

- The first constructive variant consists of: cutting with automated device, automated striping, crimping with automated device, soldering with robotic cell, manual insulation, connecting the wires with automated device and injection with automated device.
- The second constructive variant consists of: cutting with automated device, automated striping, crimping with automated device, soldering with automated device,

manual insulation, connecting the wires with automated device and injection with automated device.

- The third constructive variant consists of: cutting with automated device, automated striping, crimping with automated device, soldering with robotic cell, insulation with automated device, connecting the wires with automated device and injection with robotic cell.

As seen in Figure 22, the proposed constructive variants (constructive variant 1/2/3) are compared with the basic concept. The basic concept is represented by: cutting with automated device, manual striping, manual crimping, soldering with automated device, manual insulation, connecting with automated device and injection with automated device; this represents the current production flow in 2Connect Romania in which the results from Figure 21 were taken into account (the operations marked with the red dot were eliminated). In the first column in the figure above is the baseline that has CTQs, which are the benchmarks for comparing the constructive variants with basic concept.

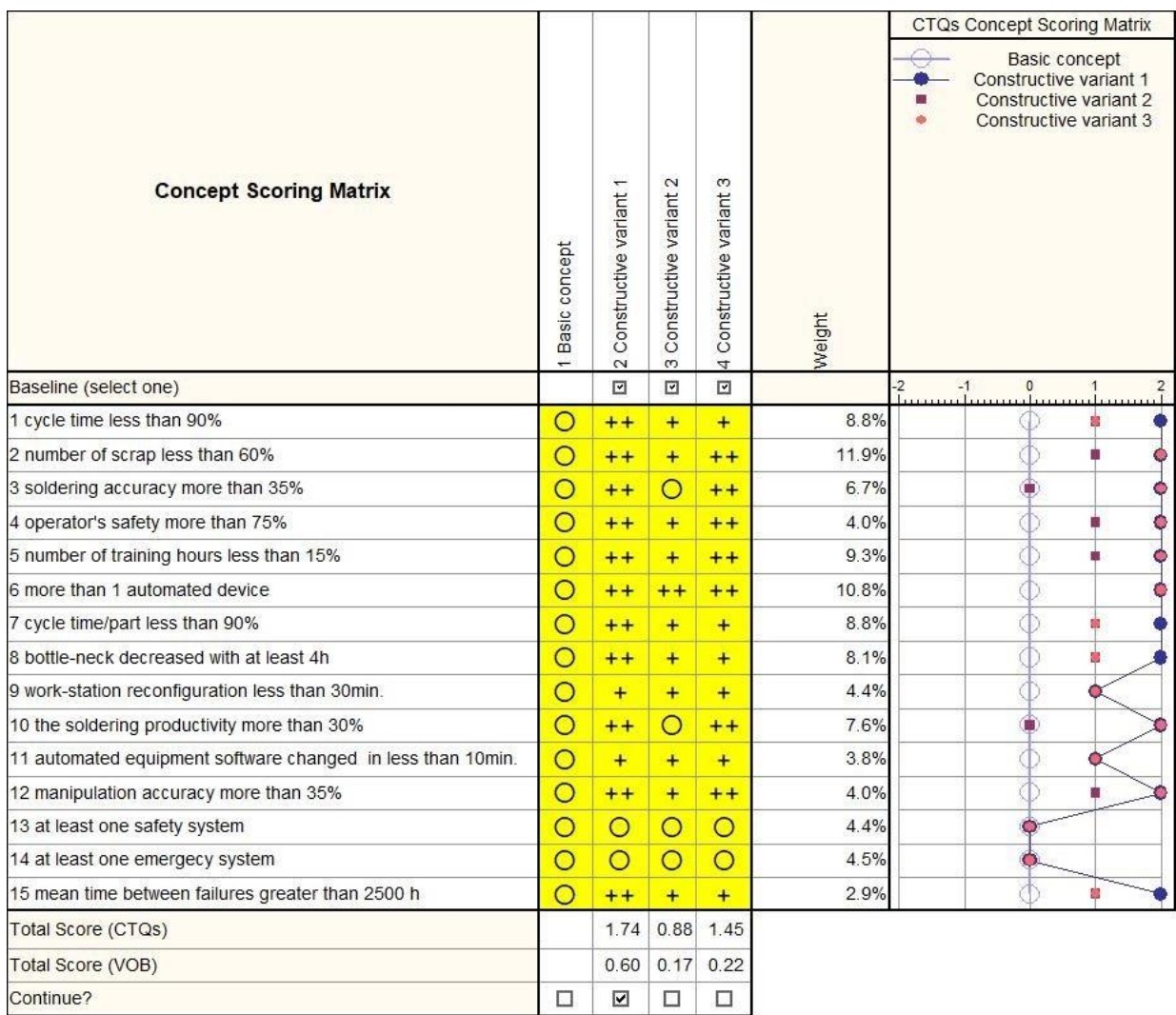

**Figure 22.** Identifying the final process flow variant.

### 4.6. Final Concept Description

Based on the information presented in previous subchapters, with reference to the equipment available in the company, the area of interest for optimization is soldering operation, implemented with a robotic cell.

The Figure 23 shows a final assembly variant of the soldering station in 2Connect Romania. The soldering station is made on an aluminum profile structure. In the lower part, you can find the command and control elements of the station and the robot. The upper part consists of two areas, the cobot's work area and the preparation area. The work area (Figure 24) is made up of a rotating table on which two trays are placed. The operator responsible for supplying the workstations stores the semi-finished products (cut cables to the established size, connectors, thermo tubes, etc.) necessary for processing the cables on the shelf and places the tray on the rotary table of the station. After the tray is assembled on the rotating table, the human operator activates the soldering process start button on the cell controller. The cell controller sends a signal to the vision camera which visually checks for reversal of wires or improperly inserted wires, and the result is transmitted as a digital signal to the robot controller acting on the scenario (moves the tray to the cobot area and starts the soldering process or stops the process and shows a warning to operator). Depending on the type of connector, after each set of pins the cable is soldered, the cobot retracts to the safe position, the work table rotates, another tray gets in the work area and the collaborative robot comes into position and the process is repeated. It is worth mentioning that the device has sensors and an integrated electronic board (for example, Arduino PCB kit, Shenzhen Wonderwing Electronics., Ltd, Shenzhen, China) that are connected to the robot controller.

In robotic applications, a very important role is played by the devices for fixing and aligning the parts in order to obtain the desired result. In this case, the device in Figure 25 was designed in order to keep the cable wires and the connector in the safe position for the cobot.

It also complies with the requirements of the flexible cell manufacturing concept, as it is composed of a modular part that is mounted and fixed to the work table and a mobile package in which the cable wires are fixed and aligned. It was chosen for the modular design because different types of connectors can be mounted on the device, this aspect being essential for the "high-mix, low-volume" production type in 2Connect Romania.

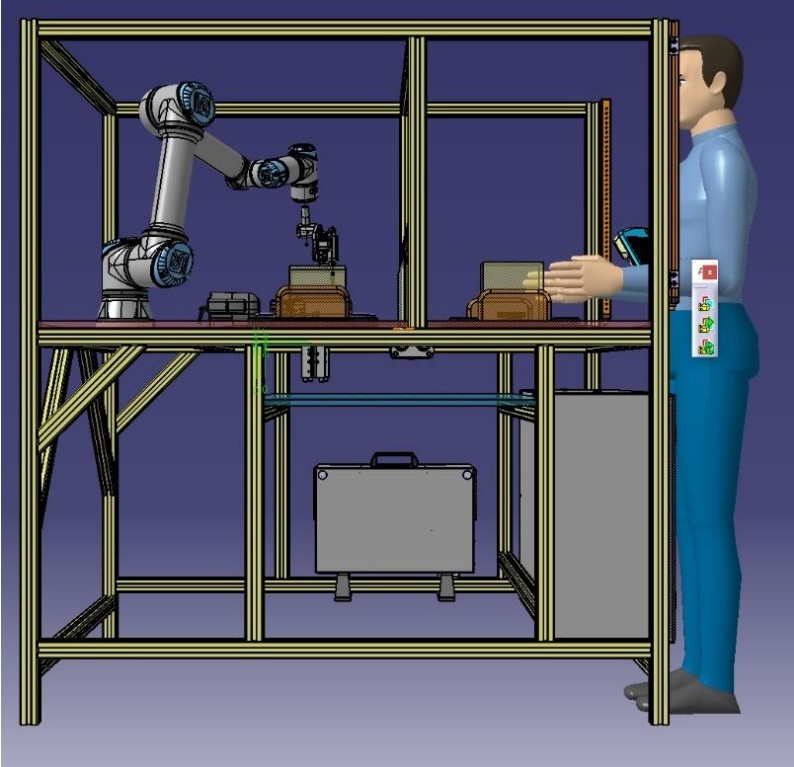

**Figure 23.** The final design of the soldering station.

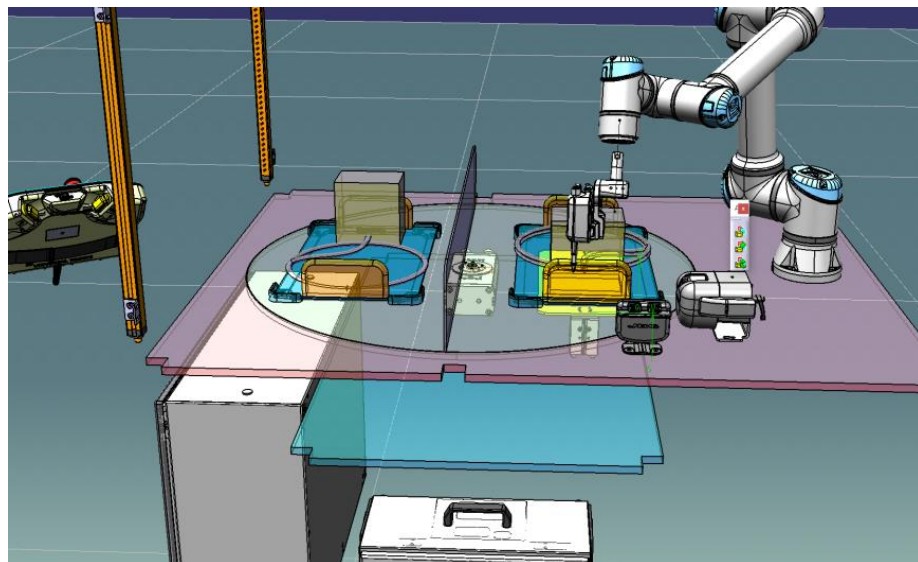

**Figure 24.** Work table of the soldering station.

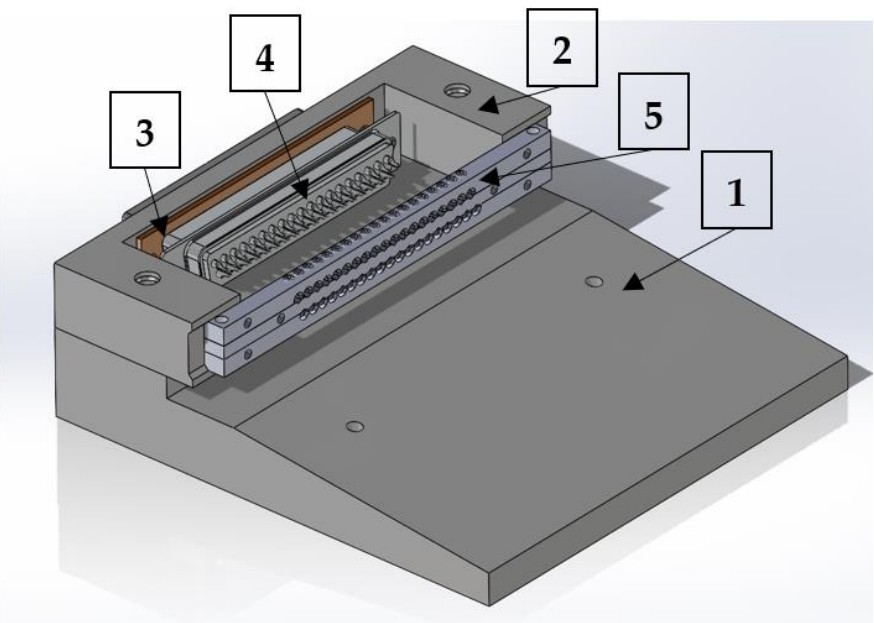

**Figure 25.** Wire fixing device.

The fixing device in Figures 25 and 26 was designed for a D-sub connector, that consists of two rows of wires, which requires the operator to flip the device to ensure soldering on both sides. The device consists of a base (1) that is mounted on the work tray by screws and centering pins. The device (2) is mounted on the base, in which the connector (4) on a gender changer (3) and the wire package (5) is mounted, which has the role of fixing and aligning the wires to be soldered in position. After the soldering operation is completed, the human operator is responsible to performed the other required operations, after removing the fixture device. Cable and connector preparation operations are carried out outside the soldering station by the human operator.

The final soldering station with all the mounted elements can be seen in the Figure 27. On the rotating table of the station, you can see the working areas and the trays with the wire fixing device.

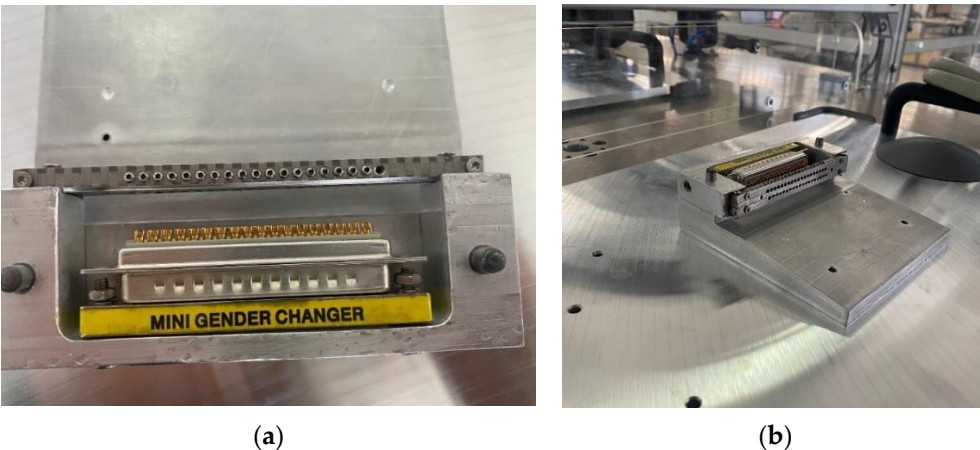

|  |  |
|:--:|:--:|
| (**a**) | (**b**) |

**Figure 26.** Wire fixing device: (**a**) gender changer and the connector; (**b**) the fixing device on the tray.

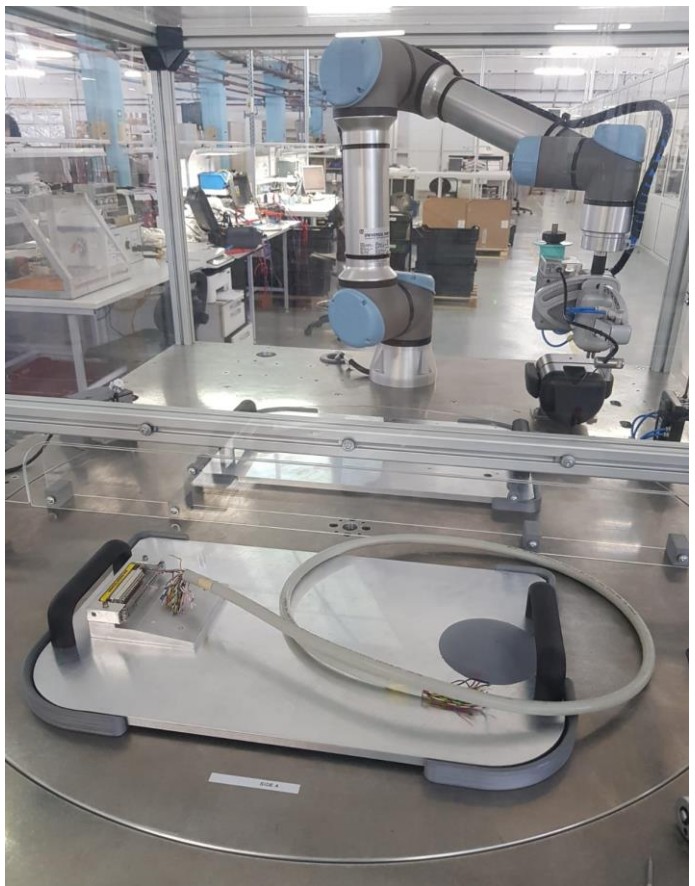

**Figure 27.** Soldering station.

## 5. Conclusions and Future Research Directions

The market of the processing of electrical/electronic cables is closely linked to the technological advance in recent years in almost all fields that have as a component digitalization, such as automotive, medical, IT, etc. That is why their production is characterized by a continuous change and, implicitly, for a company to be a competitor in this field there is a need for continuous optimization of production and the mandatory requirement of flexible manufacturing technology. This paper addressed the transition of production from 2Connect Romania to Industry 4.0 and robotization to counter the downward trend in the number of qualified employees in the industrial sector.

However, the lack of manufacturing flexibility to cope with changes in product types and their improvement, makes it apparent that the human operator may be the only one who can adapt to such uncertainties and variability. The limitations of the human worker in terms of physical strength, endurance, repeatability with low speed and high frequency of error, led to the development of research in combining the advantages of human labor with robotization automation, focusing on workstations where the robot and human operator share the same space, work and interact to accomplish tasks in the technological flow.

Using the DMADV methodology, the paper analysis the possibility of automatization the soldering process. The studied process has many limitations such as the type of process: high-mix, low-volume and the number of connectors: 37. In addition, the process is complex: the line needs to be flexible, to produce a variety of products. The standard that must be followed is: IPC/WHMA-A-620B.

Based on the studies presented in this work, the implemented soldering station improves process productivity and precision by integrating the collaborative robot in the soldering operation and smart devices for fixing and aligning the wires to be soldered.

The next step to be taken in order to develop and optimize this soldering line is to design the device for fixing and aligning the wires to be both productive (due to the large number of operations, 37 pins of the connector, for instance), but also the modularization of the device to process other types of connectors, such as to be modified easily and without much cost. This requirement is given by the type of high-mix, low-volume production.

**Author Contributions:** Conceptualization, E.P. and I.C.B.; methodology, E.C., I.C.B. and E.P.; software, D.I.; validation, I.C.B. and E.P.; resources, E.C. and I.C.B.; writing—original draft preparation, E.P. and E.C.; All authors have read and agreed to the published version of the manuscript.

**Funding:** This research received no external funding.

**Institutional Review Board Statement:** Not applicable.

**Informed Consent Statement:** Not applicable.

**Data Availability Statement:** Not applicable.

**Conflicts of Interest:** The authors declare no conflict of interest.

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
