# Peer review of "New Product Development of a Robotic Soldering Cell Using Lean Manufacturing Methodology"

_sustainability, doi:10.3390/su142114057_

Round 1
Reviewer 1 Report
Generally, the aim of the paper is attractive. Unfortunately, the article is incorrectly written, and I have some daubs about the methodology.
Consequently, I suggest the following improvements:
1) You should use the Sustainability journal template and read it carefully - e.g., what should be in each part of the text; the text should be anonymized etc.
2) In the Introduction, I suggest using some references to the literature, especially in part showing the topicality and novelty of the topic.
3) It is not enough to list publications on a given topic but to present a critical assessment of them.
4) The theoretical background is very weak. I suggest preparing an actual literature review based on a systematic literature review methodology.
5) In an international journal - like Sustainability - you shouldn't use "native references." How people who do not know a specific language should read them? - line 991.
6) You should add a paragraph (in the Literature review section) about the method you used to select and analyze the literature, for example, a systematic literature review. It would help if you informed what databases you analyzed (e.g., WoS, Scopus, ScienceDirect); what search strategy you followed. As an example of an article that is based on SLR, you can analyze:
a) Mendoza-Silva, A. Innovation capability: A systematic literature review. Eur. J. Innov. Manag. 2020. https://doi.org/10.1108/EJIM-09-2019-0263
b) Dziallas M., Blind K., Innovation indicators throughout the innovation process: An extensive literature analysis, Technovation 80–81 (2019) 3–29. https://doi.org/10.1016/j.technovation.2018.05.005
7) The research gap - should precisely result from the literature review.
8) What about the hypotheses or research questions? You should precisely present them. The hypothesis or RQ should be developed based on the literature review. I suggest the sub-section - "hypothesis development."
9) The theoretical/conceptual research model should be prepared after SLR. It should contain hypotheses or research questions.
10) Theoretical part does not contribute anything. It looks like a textbook.
11) You should use a classic version of the article structure. In each part, you should present the critical information or data. For example, in the material and methods part, you give the method (e.g., correlation, case study, qualitative or quantitative) and materials (data) you used. It's entirely not clear in your text.
12) In conclusion, you should present the main results, research limitations, and further research directions.
Author Response
Please see the attachment and the revised paper.

Reviewer 2 Report
The paper presents the application of the product design methodology for the development of an automated soldering cell for electronic cables. Process developed very interesting and prospective. In general, the structure as the paper is presented leaves the reader confused! A restructuring could be carried out (introduction, methodology, results and conclusions). More objectivity is also expected in the text, that is, a lot of information is repeated in the text. In my opinion, a good review of the article is necessary to be accepted. Other considerations were made in the attached file.

Author Response

(The authors gave the same response as above.)

Reviewer 3 Report
Page 5 line 252
In [33] the critical to quality characteristics needed for the integration of a cobot with a CNC machine was study.
Check grammar of the sentence, something is wrong
On page 9 line 376
view.(for this sector the manufacturing type: high mix low volume is representative).
You have an extra dot
I read the paper and it was really interesting first the technological part and then how you used all quality instruments to identify problems. Don’t get me wrong, the paper is really interesting but there is no scientific contribution and the paper is too long. Either you concentrate on the first part which is highly technical, or you concentrate on quality tools and how the help improve quality with your case study. I also did not understand did you file for a patent and you are describing it here or you citing someone’s research. You have to be clearer on that.
Author Response

(The authors gave the same response as above.)

Round 2
Reviewer 1 Report
Dear Authors,
I'm glad you found my suggestions helpful. I believe that the article can be published in this form.
Reviewer 2 Report
The authors made the changes suggested in the paper. Therefore, I consider the paper suitable for publication.
Reviewer 3 Report
The authors really put a great effort to clarify the paper and put a greater emphasis on quality tools. I think now the paper is adequately enhanced to merit publishing.